# Integrated Multi-Tumor Radio-Genomic Marker of Outcomes in Patients with High Serous Ovarian Carcinoma

**DOI:** 10.3390/cancers12113403

**Published:** 2020-11-17

**Authors:** Harini Veeraraghavan, Herbert Alberto Vargas, Alejandro Jimenez-Sanchez, Maura Micco, Eralda Mema, Yulia Lakhman, Mireia Crispin-Ortuzar, Erich P. Huang, Douglas A. Levine, Rachel N. Grisham, Nadeem Abu-Rustum, Joseph O. Deasy, Alexandra Snyder, Martin L. Miller, James D. Brenton, Evis Sala

**Affiliations:** 1Department of Medical Physics, Memorial Sloan Kettering Cancer Center, New York, NY 10065, USA; deasyJ@mskcc.org; 2Department of Radiology, Memorial Sloan Kettering Cancer Center, New York, NY 10065, USA; vargasah@mskcc.org (H.A.V.); lakhmany@mskcc.org (Y.L.); es220@medschl.cam.ac.uk (E.S.); 3Cancer Research UK Cambridge Institute, University of Cambridge, Li Ka Shing Center, Cambridge, Cambridgeshire CB2 0RE, UK; Alejandro.JimenezSanchez@cruk.cam.ac.uk (A.J.-S.); mireia.crispinortuzar@cruk.cam.ac.uk (M.C.-O.); martin.miller@cruk.cam.ac.uk (M.L.M.); James.Brenton@cruk.cam.ac.uk (J.D.B.); 4Radioterapia Oncologica ed Ematologica, Dipartimento Diagnostica per Immagini, Area Diagnostica per Immagini, Radiologica Diagnostica e Interventistica Generale, Fondazione Policlinico Universitario A. Gemelli IRCCS, 00168 Roma, Italy; miccomaura@yahoo.it; 5Columbia University Medical Center, New York, NY 10032, USA; eralda.mema@gmail.com; 6National Cancer Institute, Rockville, MD 20850, USA; erich.huang@nih.gov; 7Laura and Issac Perlmutter Cancer Center, New York University Langone Health, New York, NY 10016, USA; douglas.levine@nyulangone.org; 8Department of Medicine, Memorial Sloan Kettering Cancer Center, New York, NY 10065, USA; grishamr@mskcc.org (R.N.G.); snyderalex@gmail.com (A.S.); 9Department of Medicine, Weill Cornell Medical College, New York, NY 10065, USA; 10Department of Surgery, Memorial Sloan Kettering Cancer Center, New York, NY 10065, USA; abu-rusn@mskcc.org

**Keywords:** machine learning, radiomics, high grade serous ovarian cancer, computed tomography, chemotherapy response prognostication, intra-site and inter-site radiomic heterogeneity

## Abstract

**Simple Summary:**

Clinical responses to the initial treatment of high grade serous ovarian cancer (HGSOC) vary greatly. Widespread intra-site and inter-site genomic heterogeneity presents significant challenges for the development of predictive biomarkers based on pre-treatment sampling of select individual tumors. Non-invasive stratification of patients with HGSOC by risk of outcome could facilitate a higher level of intervention for those with the highest risk of a poor outcome. We developed and validated a machine learning-based integrated marker of HGSOC outcomes to standard chemotherapy that combines a previously developed intra-site and inter-site CT radiomics measure called cluster dissimilarity (cluDiss) with clinical and genomic measures using two retrospective cohorts of internal and external institution datasets. Our approach was more accurate than conventional clinical and average radiomics measures for prognosticating progression-free survival and platinum resistance.

**Abstract:**

*Purpose:* Develop an integrated intra-site and inter-site radiomics-clinical-genomic marker of high grade serous ovarian cancer (HGSOC) outcomes and explore the biological basis of radiomics with respect to molecular signaling pathways and the tumor microenvironment (TME). *Method:* Seventy-five stage III-IV HGSOC patients from internal (*N* = 40) and external factors via the Cancer Imaging Archive (TCGA) (*N* = 35) with pre-operative contrast enhanced CT, attempted primary cytoreduction, at least two disease sites, and molecular analysis performed within TCGA were retrospectively analyzed. An intra-site and inter-site radiomics (cluDiss) measure was combined with clinical-genomic variables (iRCG) and compared against conventional (volume and number of sites) and average radiomics (*N* = 75) for prognosticating progression-free survival (PFS) and platinum resistance. Correlation with molecular signaling and TME derived using a single sample gene set enrichment that was measured. *Results:* The iRCG model had the best platinum resistance classification accuracy (AUROC of 0.78 [95% CI 0.77 to 0.80]). CluDiss was associated with PFS (HR 1.03 [95% CI: 1.01 to 1.05], *p* = 0.002), negatively correlated with *Wnt* signaling, and positively to immune TME. *Conclusions:* CluDiss and the iRCG prognosticated HGSOC outcomes better than conventional and average radiomic measures and could better stratify patient outcomes if validated on larger multi-center trials.

## 1. Introduction

Ovarian cancer accounts for approximately 239,000 new cases and 152,000 deaths worldwide annually [1]. High grade serous ovarian carcinoma (HGSOC) is the deadliest gynecologic malignancy and is associated with a very poor prognosis [2]. Although HGSOC shows marked sensitivity to initial platinum-based chemotherapy [3], most patients recur and become progressively resistant to subsequent treatments [4]. Acquisition of resistance may be related to specific mutational processes that drive genomic heterogeneity [5,6] and clonal evolution [7,8]. HGSOC exhibits marked intra-site and inter-site genomic heterogeneity across metastatic sites in the peritoneal cavity [6,7,8] with altered immunological infiltrates and a tumor micro-environment (TME) [9]. Detection of spatial or temporal heterogeneity by multiple sampling in a single patient is expensive, invasive, and often clinically impractical. Consequently, analysis of heterogeneity has only been performed as retrospective research studies on a limited number of patients with HGSOC [6,7,8]. There is a pressing need for facile and non-invasive measures for intra-site and inter-site radiomic heterogeneity that can be integrated into clinical pathways.

Computed tomography (CT) and serum CA-125 measurement are routinely used for the initial staging and treatment monitoring of patients with HGSOC, but standard imaging protocols do not provide information on tumor heterogeneity. Texture analysis of CT data is a radiomics method [10,11] that can provide detailed quantitative characterization of local variations in intensity levels throughout an image. The majority of radiomics methods compute average measures of tumor heterogeneity based on a single site of disease per patient even in those with metastatic disease [10,12,13,14,15,16,17,18,19], including a recent study of patients with advanced ovarian cancer from preoperative CT images [12]. However, averaged radiomics measures do not capture the potential variability within different regions of a tumor and between multiple tumors in the same patient.

Prior studies by our group have demonstrated that radiomic features quantifying the heterogeneity between tumor sites are associated with shorter overall survival (OS) and incomplete surgical resection in HGSOC patients treated with chemotherapy [20] as well as with shorter progression-free survival (PFS) in a different cohort of HGSOC patients with BRCA1/2 mutation [21]. More recently, we extended these methods to incorporate both intra-site and inter-site radiomic heterogeneity (IISH) and showed that a single measure, known as cluster dissimilarity (cluDiss), was associated with an immunotherapy response in patients with recurrent HGSOC [22]. These results show that modeling radiomic heterogeneity between the different sites of disease can help to better stratify patients with HGSOC.

In this retrospective study, we validated the cluDiss marker to stratify outcomes in HGSOC patients before chemotherapy treatment. Furthermore, we developed an integrated marker combining intra-site and inter-site radiomics-clinical-genomic (iRCG) variables using machine learning to distinguish patients’ outcomes. The aims of this study were to (i) validate cluDiss as a predictor of outcomes using an internal and external multi-institutional cohort, and (ii) evaluate whether an integrated iRCG measure of HGSOC outcomes was more accurate than average heterogeneity radiomic (aRCG) and conventional imaging (CCG) measures. Finally, we attempted to establish the biological basis of the prognostic radiomics measures by studying their correlation with underlying biological processes characterized by a well-defined molecular HALLMARK gene set pathways, stromal and immune scores of the tumor microenvironment (TME), and established 18 cell types of the TME extracted using the consensus^TME^ method [23,24] by using patient-level single sample gene set analysis (ssgsea) [25] from RNA-sequencing data.

## 2. Results

### 2.1. Patient and Tumor Characteristics

The REMARK diagram flowchart for selecting the patients is described in Appendix A. The patient clinical characteristics are shown in Table 1. The median follow-up was 41.9 mos (inter-quartile range [IQR] 22.9 months [mos]—56.3 mos) in the internal Memorial Sloan Kettering Cancer Center (MSKCC) cohort and 19.3 mos (IQR 6.3 mos—38.6 mos) in the Cancer Imaging Archive (TCIA) cohort. All but two patients in MSKCC and 17 patients in TCIA experienced progression during the follow-up period. The median number of tumor sites was 7 (IQR 6 to 9) for the MSKCC and 4 (IQR of 3 to 5) for the TCIA cohort. 

In total, 460 tumor volumes of interest (VOI) were analyzed to compute the intra-site and inter-site tumor heterogeneity (IITH) metric cluster dissimilarity (cluDiss) as well as several (*N* = 75) average heterogeneity radiomics measures. The total tumor burden volume (TTV) was 122.0 cc (IQR of 65.5 cc to 229.0 cc) in MSKCC and 331.0 cc (IQR of 158.2 cc to 595.0 cc) in the TCIA datasets. After excluding 14 patients who had no platinum resistance data, 61 patients were analyzed for platinum resistance classification. Forty-four cases had matched imaging and RNA-sequencing data and were used for radio-genomic analysis.

### 2.2. Association with Survival (PFS)

The intra-inter site tumor heterogeneity radiomics-clinical-genomics (iRCG) iRCG_PFS_ score was computed using the best tuning parameters (α = 1, λ = 0.826), as (1).
(1)$${\mathrm{iRCG}}_{\mathrm{PFS}}=4.44\times \mathrm{cluDiss}+3.72\;\times \;\mathrm{age}+2.11\times \;\mathrm{CNB}\;$$

The conventional-clinical-genomic (CCG) CCG_PFS_ score was computed with best tuning parameters of α = 1, λ = 0.809, as (2)
(2)$${\mathrm{CCG}}_{\mathrm{PFS}}=2.86\times \mathrm{age}+1.85\times \mathrm{sites}+1.17\times \mathrm{CNB}+0.235\times \mathrm{resection}\;$$

The average radiomic-clinical-genomic (aRCG) aRCG_PFS_ score was computed with (α = 1, λ = 0.803), as (3):

(3)$$\begin{array}{ll} {\mathrm{aRCG}}_{\mathrm{PFS}}=2.08 & \times \mathrm{age}+1.30\times \mathrm{CNB}+1.22\times \mathrm{resection}+0.10\times \mathrm{SZN}+1.26\\ & \times \mathrm{coarseness}+1.73\times \mathrm{Sobel}.\mathrm{mean}+0.30\\ & \times \mathrm{Gabor}\left(45^{{}^{\circ}},\;\sqrt{2}\right).\mathrm{kurtosis}+1.41\times \mathrm{Gabor}\left(90^{{}^{\circ}},\;\sqrt{2}\right).\mathrm{mean}+0.30\\ & \times \mathrm{Gabor}\left(90^{{}^{\circ}},\;\sqrt{2}\right).\mathrm{skewness}+1.91\times \mathrm{Gabor}\left(135^{{}^{\circ}},\;\sqrt{2}\right) \end{array}$$

Both cluDiss and continuous iRCG_PFS_ scores were associated with PFS in both univariate and multivariable analysis (after adjusting for clinical factors and copy number burden [CNB]). Total tumor volume (TTV), CCG_PFS_, and aRCG_PFS_ scores were not associated with progression-free survival (PFS) (Table 2). The number of sites was associated with PFS in both univariate and multivariable analysis for the TCIA and to PFS in the univariate model in the MSKCC dataset. These results showed that the cluDiss measure was able to stratify patients by PFS better than both conventional imaging and average heterogeneity radiomics measures. 

The cluDiss measure achieved a concordance probability estimate (CPE) for PFS of 0.66 (95% CI of 0.61 to 0.70) for MSKCC and 0.67 (95% CI of 0.63 to 0.72) for TCIA cohorts. The CPE for the number of sites was (MSKCC CPE of 0.59 [95% CI of 0.55 to 0.64], TCIA CPE of 0.655 [95% CI of 0.54 to 0.77]), and TTV was (MSKCC CPE of 0.52 [95% CI of 0.47 to 0.57], TCIA CPE of 0.535 [95% CI of 0.46 to 0.61]), respectively.

The iRCG_PFS_ score that combined cluDiss with clinical (age, state, and resection status) and genomic CNB measure achieved a CPE of 0.69 (95% CI of 0.65 to 0.74) for MSKCC and 0.695 (95% CI of 0.64 to 0.75) for TCIA cohorts, respectively. On the other hand, both CCG_PFS_ (CPE: MSKCC 0.60 [95% CI 0.55 to 0.65], TCIA 0.54 [95% CI 0.48 to 0.61]), aRCG_PFS_ (CPE: MSKCC 0.53 [95% CI 0.49 to 0.58], and TCIA 0.51 [95% CI 0.44 to 0.58]) had lower CPEs. The iRCG_PFS_ produced the highest CPE of all other integrated models. It was slightly better than the cluDiss measure alone. 

The iRCG_PFS_ and cluDiss cut-off to dichotomize patients into high risk (≥cut-off) and low risk (<cut-off) were determined as 642.00 and 68.62, respectively, on the MSKCC dataset. Testing on the TCIA dataset with this same cut-off showed significantly longer PFS for the lower values of iRCG_PFS_ (*p* = 0.0006) and lower values of cluDiss (*p* < 0.001), respectively (Figure 1).

### 2.3. Classification of Platinum Resistance

The iRCG linear SVM model (AUROC of 0.78, 95% CI 0.77 to 0.79) was significantly more accurate than CCG linear SVM (*p* < 0.001) and aRCG RFE-SVM (*p* = 0.004) (Table 3). The receiver operating characteristic curves for all three classifiers are shown in Figure 2. The iRCG SVM had the highest sensitivity of the three methods for classifying patients likely to develop platinum resistance.

All variables except CNB were relevant (importance > 0) in the iRCG and CCG models. Resection status was the most relevant feature for all three models (Importance score = 100), while cluDiss had a lower importance score of 34.4, clearly indicating the relevance of the clinical variables for predicting platinum resistance. Two radiomic measures, dependence counts non-uniformity (DCN), and the gray level non-uniformity (GLN) were found to be relevant in the aRCG model. CNB had a low importance score of 2.32 in the aRCG model. 

Improved prognostication of HGSOC outcomes using the iRCG measures could allow for better upfront and non-invasirve strafication of patients with HGSOC by risk of outcome than with the conventional clinical or genomic measures alone. More accurate risk stratification could faciliate a higher level of intervention for those with the highest risk. 

### 2.4. Robustness to the CT Scanner Manufacturer

The cluDiss measure did not show statistical difference between scanners (*p* = 0.06) (Appendix A). Of the 75 average radiomic measures, 24 were robust to scanner differences and these same radiomic featuers were used in constructing the integrated aRCG models of PFS and platinum resistance. Note that the iRCG model only used cluDiss as the radiomic measure. Four out of 13 gray level run length matrix (GLRLM) (30.8%), 5 out of 13 gray level size zone matrix (GLSZM) (38.5%), 2 out of 5 neighborhood gray tone difference matrix (NGTDM) (40%), 4 out of 14 neighborhood gray level difference matrix (NGLDM) (28.6%), and 9 out of 20 edge features (45%) did not show statistical differences while both first order and gray level correlation matrix (GLCM) measures showed significant differences between scanners. Gray level non-uniformity (ρ = 0.717) and dependence count non-uniformity (DCN) (ρ = 0.606) were highly correlated with TTV. The cluDiss feature, which is designed to increase with the number of lesions, was highly correlated with the number of sites (ρ = 0.833) (Appendix A).

### 2.5. Correlation of Cludiss to Biological Processes

We studied the differences in the molecular signaling pathways between the low-risk and high-risk patient groups using the 50 hallmark gene sets [26], extracted using single sample gene-set enrichment (ssgsea) analysis of the RNA samples [25]. The signaling pathways were categorized into immune, oncogenic, stromal, cellular, and other [24]. Patients were dichotimized using the median value (cluDiss = 68.6) of cluDiss (low-risk<median and vice versa). Principal component analysis (PCA) of the 50 gene sets showed that the *MYC, MTORC1* pathways were relevant in the high-risk group but not in the low risk group (Figure 3) in the first two PC dimensions (>60% variation explained). The 50 gene set expression variations for both groups (Appendix A) showed many gene-sets contributing to patient variability for the high-risk (in the first two dimensions) compared to the low-risk group.

The Spearman rank correlation of the features cluDiss, DCN, and GLN that were relevant for platinum resistance classification were measured against stromal and immune scores computed using the ESTIMATE method [27], the 50 hallmark gene sets [26], and the 18 consense^TME^ cell types [23,24]. TTV and sites were also evaluated for correlation for completeness. Because of the reported widespread TME heterogeneity in HGSOC patients both within and between tumor sites [7,24], we computed two variations of the cluDiss measure using only the tumor sites in the pelvis and only the tumors in the abdominal sites to assess their correlation with the biological processes. 

CluDiss computed from across all visible tumor sites was negatively correlated with the *Wnt* signaling pathway (ρ= −0.35, *p* = 0.02). CluDiss measure from the pelvic sites showed no significant correlation to any of the gene set pathways. CluDiss measure computed from the abdominal metastases was negatively correlated with *Wnt* and *NOTCH* signaling (Figure 4, Appendix A), positively correlated with *MTORC1*, and allograft rejection (Figure 4), immune cells (ρ= 0.33, *p* = 0.028). This same measure was positively correlated to 13 of the 18 TME cell types (Figure 4, Appendix A), including Tgd (ρ= 0.40, *p* = 0.007), Treg (ρ= 0.39, *p* = 0.009), Bcells (ρ= 0.39, *p* = 0.008), CD4 (ρ= 0.41, *p* = 0.006), CD8 (ρ= 0.40, *p* = 0.007), cytotoxic (ρ= 0.44, *p* = 0.003), and NK (ρ= 0.43, *p* = 0.004) cell types. DCN and GLN both measure the heterogeneity in the distribution of the local signal intensities, which are highly correlated (ρ= 0.98, *p* < 0.0001) with each other. DCN was negatively correlated with immune gene-sets (Figure 4) (Appendix A), immune cells (ρ= −0.32, *p* = 0.033), and 11 of the TME cell types (Figure 4, Appendix A), including Tgd (ρ= −0.46, *p* = 0.002), Treg (ρ= −0.42, *p* = 0.005), B cells (ρ= −0.41, *p* = 0.006), CD4 (ρ= −0.41, *p* = 0.005), CD8 (ρ= −0.42, *p* = 0.004), cytotoxic (ρ= −0.34, *p* = 0.02), and NK (ρ= −0.45, *p* = 0.002) cell types.

In short, the gene set expressions were different between the low and high-risk patient groups dichotomized using the cluDiss measure. Furthermore, cluDiss, which quantifies the textural heterogeneity between multiple tumor sites, was positively correlated with the immune cell type and negatively with *Wnt* signaling pathway, while the average texture heterogeneity measures known as DCN and GLN showed a negative correlation with immune gene sets and immune cell types. Prior work by our group [24] has shown that enrichment of *Wnt* and *Myc* is negatively correlated with immune infiltration. 

## 3. Discussion

Non-invasive stratification of patients with HGSOC by risk of outcome could facilitate a higher level of intervention for those with the highest risk of poor outcome. Possible therapeutic/diagnostic interventions could include enrollment in clinical trials, addition of bevacizumab to first line chemotherapy, and more frequent follow-up imaging to evaluate progression. Building on prior reports [12] that highlight the relevance of radiomic measures for predicting HGSOC outcomes, we validated a novel IISH measure, cluDiss, that we previously showed to be associated with HGSOC outcomes in a different cohort of patients [22]. Unlike most radiomic studies [10,12,16] that compute an average measure of single tumor heterogeneity, cluDiss quantifies the heterogeneity within and between the entire tumor burden rather than just the ovarian mass. Its magnitude increases with the number of sites and the textural differences between the sub-regions within and between the tumor sites, reflecting larger imaging heterogeneity. It adds to the conventional number of sites measured by quantifying the radiomic heterogeneity in those lesions.

The cluDiss measure did not show significant differences to CT scanners, as did 24 of the 75 average heterogeneity measures. However, both first-order histogram and all GLCM measures showed a significant difference between scanners.

Quantifying inter-site and intra-site imaging heterogeneity is important because HGSOC exhibits widespread genomic intra-site and inter-site heterogeneity. Multi-site genomic studies have shown intratumor genomic heterogeneity to correlate to poor survival [7]. In addition to clonal heterogeneity, the malignant cell spread within the peritoneal cavity is distinct and non-random [8,9], as some sites harbor genetically diverse clones [8]. These site-specific properties, including immunologic components of the TME, may modulate malignant cell invasion and expansion, thereby shaping evolutionary selection [28]. However, large scale multi-site genomic heterogeneity studies are difficult to do and are impractical for clinical practice. This motivated the development and validation of a non-invasive CT-based measure of intra-site and inter-site radiomic heterogeneity. 

More importantly, our results show that the integrated model combining cluDiss, clinical, and genomic variables was more accurate than the conventional imaging (total tumor volume and number of sites) and average tumor heterogeneity radiomics models for predicting PFS. This finding is in line with other works that have shown integrated radio-genomic models to better predict outcomes in other solid cancers [29,30]. Although cluDiss was as good as the iRCG_PFS_ measure for predicting PFS, the platinum resistance classification benefitted from the clinical measures, indicated by their higher importance over cluDiss for that model. On the other hand, CNB, while relevant for the iRCG_PFS_ model, was not relevant for classifying platinum resistance. Both cluDiss and the clinical measures can be obtained in a non-invasive way and their combination could, thus, serve to non-invasively stratify patient outcomes without needing genomic measures. On the other hand, genomic measures could be used for obtaining a mechanistic drivers of risk in those patients determined to be high risk using the non-invasive measures, such as by finding activated or suppressed pathways. 

Understanding the radio-genomic correlations are important for furthering their application as biomarkers of treatment response [11]. Multiple studies have reported the association of image-based qualitative [31] and quantitative radiomics features with genomic measures [32,33,34,35,36]. We measured the correlation of cluDiss and two radiomics measures known as DCN and GLN with the Hallmark gene sets and TME cell types extracted from ssgsea analysis of the RNA expressions. Gene sets are candidates for genes that may either drive genomic heterogeneity or are required for survival in the context of ongoing chromosomal instability. Patients dichotomized into low-risk and high-risk groups using cluDiss showed a difference in the gene set pathways. Additionally, cluDiss was negatively correlated with the *Wnt* signaling pathway and positively to 13 immune TME cell types including Tregs, Tgd, Bcells, CD4, CD8, and NK. Prior work by our group [24] showed a mutual exclusivity in the expression of *Wnt* and *Myc* gene pathways with respect to the immune cell types in untreated HGSOC patients. CluDiss has also been shown to correlate with *CKB* protein in a previous study using matched imaging and proteomic samples from 20 HGSOC patients by our group [37]. To our knowledge, this is the first report on the radio-genomic correlation of intra-site and inter-site radiomic heterogeneity in HGSOC. 

Our study is limited by a small dataset with high class imbalance (e.g., higher prevalence of platinum sensitivity than resistance), which was partly mitigated through a synthetic minority oversampling technique [17] by using a linear SVM classifier. Additionally, the genomic samples were available from only a single primary tumor site. Hence, a study of variability in gene sets and CNB between tumor sites, or their impact on classifying outcomes was not possible. Also, a study of repeatability of the radiomic measures needs to be assessed with test-retest studies and under different CT acquisition protocols. Studying the potential for clinical translation would require larger multi-institutional cohorts.

## 4. Materials and Methods

### 4.1. Ethics and Consent 

The Institutional Review Board approved this retrospective Health Insurance Portability and Accountability Act-compliant study and waived the requirement for written informed consent. The TCIA is a managed, publicly available, open-source repository of de-identified medical images of cancer and corresponding patient information that is sourced from 28 participating institutions [38].

### 4.2. Study Design and Patients

Two cohorts of patients with HGSOC: a single institution dataset from MSKCC (*N* = 45) and a multi-institution dataset (*N* = 38) from the ovarian-TCIA [31], which included patients treated at five different institutions (Appendix A) were identified from which 75 patients were selected. The eligibility criteria included: (i) federation of international gynecologic oncology (FIGO) stage III-IV HGSOC, (ii) attempted primary cytoreductive surgery, (iii) standard of care contrast-enhanced CT of the abdomen and pelvis performed prior to surgery, (iv) at least two tumor sites identified on CT for computing cluDiss, and (v) molecular analysis performed within The Cancer Genome Atlas (TCGA) Research Network ovarian cancer pilot project. Patients who received neoadjuvant chemotherapy prior to surgery, five patients from MSKCC, one from TCIA who did not complete molecular analysis, and two from TCIA who did not have data regarding surgical resection were excluded. Sixty-seven patient scans (89%) were acquired with voltage 120 kVp (median 120, IQR 110 to 120), tube current (median 300 mA, IQR of 89 mA to 393 mA), and reconstructed with a standard convolutional kernel with the most common slice thickness of 5 mm for 54 (72%) patients (median 5 mm, IQR 2 mm to 5 mm).

All patients used in this study were previously used for qualitative radiologist CT assessments based on association with Classification of Ovarian Cancer transcriptomic profiles and survival [31,39]. Thirty-eight patients from the MSKCC dataset were used with inter-site heterogeneity measures for predicting OS and surgical resection status [20]. The cluDiss measure, developed in our prior work on an entirely different cohort of patients for analyzing response to immunotherapy treatment in HGSOC [22], combines both intra-site and inter-site radiomic heterogeneity. 

Our study aims (Figure 5) were to: (a) evaluate association of cluDiss and iRCG with PFS, (b) compare iRCG classifier of platinum resistance against models combining clinical and genomic variables with conventional imaging (tumor volume, number of sites) (CCG), and average heterogeneity radiomic (*N* = 75) features (aRCG). We evaluated the robustness of cluDiss and average heterogeneity radiomic measures against CT scan manufacturers. Finally, we explored the biological basis of these measures on a subset of patients (*N* = 44) with matched CT imaging and molecular RNA-sequencing data by measuring correlation with well-defined Hallmark gene set signaling pathways, an immune and stromal tumor micro-environment (TME), and 18 TME cell types.

Clinical variables such as patient age, FIGO stage, and cytoreductive outcome (complete gross resection, optimal (≤1 cm residual disease), or suboptimal resection (>1 cm residual disease)) were obtained from patient clinical records for the internal MSKCC dataset and were available through cbioportal [40] for the TCIA dataset.

The copy number burden (CNB) was computed as the fraction of the altered genome and is available from the cbioportal [40] for outcome classification. The CNB is a measure of genome instability and is computed as the length of segments (in log2 scale) with copy number alterations >0.2 and divided by the length of the measured segments [40]. CNB was computed from the genomic sample taken from a single primary tumor site using common guidelines in the TCGA ovarian cancer study [41]. Specifically, biospecimens were collected from newly diagnosed ovarian cancer serous adenocarcinoma patients undergoing surgery. One tumor and matched normal tissue specimen were collected for each patient. 

Platinum resistance was defined as a platinum-free interval of less than 6 months after initial therapy [42]. PFS was calculated as the time from the date of primary surgery to the date of documented first recurrence on the basis of findings on a CT scan, physical examination results, or death prior to recurrence. 

### 4.3. Computation of Intra-Site and Inter-Site Tumor Radiomic Heterogeneity

The IISH cluster dissimilarity (cluDiss) measure was computed as follows.
All suspected primary and metastatic tumors in the abdomen and pelvis (>1 cm) were manually delineated by two oncologic imaging research fellows (4 and 6 years of experience, respectively) using 3DSlicer [43], thereby resulting in multiple volumes of interest (VOI). Two conventional imaging measures, total tumor volume (TTV), estimated as the total number of voxels within each VOI multiplied by the voxel size, and the number of anatomic sites corresponding to the number of radiologist-defined sites of disease on preoperative CT scans were computed.CT images were rescaled to 0-255 and discretized into 32 bins. Then, Haralick textures, energy, entropy, homogeneity, and contrast were computed [20] by sliding a fixed sized patch (11 × 11 × 1) centered around every voxel within all VOIs using in-house software implemented in C++ using the Insight ToolKit (ITK) [44].Sub-regions of homogeneous texture were extracted from within VOIs by grouping voxels with similar texture values using kernel K-means clustering [45], which exploits the spatial relatedness of voxels to produce a computationally fast clustering. The appropriate number of clusters for each patient was determined using Akaike information criterion from an empirically chosen maximum of five clusters. The mean values of the four individual Haralick texture measures described the sub-regions.Sub-region textural differences were quantified using Euclidean distance and summarized into a dissimilarity matrix.The group dissimilarity matrix (GDM), which is a 2D histogram that captures the number of sub-region pairs with similar levels of dissimilarity, was computed. The rows of the GDM correspond to the number of sub-regions with a similar dissimilarity and the columns correspond to the dissimilarity level. Ten bins were used to discretize the dissimilarities and the sub-region pairs sizes following min-max normalization.The cluDiss measure, which quantifies the peakedness in the distribution of dissimilarities by considering the relatedness between groups of subregions by sharing similar levels of dissimilarity within the GDM is computed as:
(4)$$\mathrm{cludiss}=\sqrt{\frac{1}{K\times M}\overset{K}{\underset{i}{\sum }}\overset{M}{\underset{j}{\sum }}{\left(i+j-\mu D-\mu A\right)}^{4}\times G\left(i,j\right),\;}$$
where K is the number of dissimilarity levels and M is group size levels, μD and μA are the normalized mean of dissimilarity levels and group sizes, and G is the group dissimilarity matrix. The indicies i and j emphasize larger dissimilarities and larger group sizes. Higher cluDiss values result from the presence of many texturally distinct sub-regions (Appendix A), while fewer large texturally distinct groups with distinct dissimilarity will result in lower cluDiss values (Appendix A).

### 4.4. Computation of Average Radiomic Heterogeneity Measures

Average heterogeneity radiomic texture measures (*N* = 75) (Appendix A) quantifying the textural heterogeneity across all disease sites were computed using the computational environment for radiological research (CERR) [46] software (https://github.com/cerr/CERR/). Shape metrics were not computed because they only quantify characteristics of single tumors. Extracted features included a first order histogram (*N* = 4), a gray level correlation matrix, GLCM (*N* = 5), a gray level run length matrix, GLRLM (*N* = 13), a gray level size zone matrix, GLSZM (*N* = 13), a neighborhood gray tone difference matrix, NGTDM (*N* = 5), a neighborhood gray level dependence matrix, NGLDM (*N* = 15), mean values of Sobel (*N* = 4), and Gabor edges at orientations ($0^{{}^{\circ}}$, $45^{{}^{\circ}}$, $90^{{}^{\circ}}$, $135^{{}^{\circ}}$), and a bandwidth of √2 (16 features). All of the radiomic features were compliant with the imaging biomarker standardization initiate (IBSI) [47] and default parameter settings available in CERR that were tested for IBSI compliance were used. 

### 4.5. Single Sample Gene Set Enrichment Analysis

Single sample gene set enrichment analysis (ssgsea) [25] was performed on the RNA measurements of each sample using the *GSVA* package version 1.28.0 in R version 3.5.0 [25]. Default settings of parameter τ=0.25 as originally used in Reference [48] was employed to place a modest weight on the expression of genes in a gene set pathway. This parameter corresponds to the weight associated with the ranking of absolute expression of genes in a signature of pathway in relation to the expression of all other genes. Normalized enrichment scores were then generated and combined with the gene ontology MSigDB [26] to estimate the pathway enrichment for the 50 Hallmark gene sets. The estimation of stromal and immune cells in malignant tumor tissues using expression data (ESTIMATE) method [27] was used to quantify the immune and stromal signatures from the bulk tumor RNA-sequence data. The results of ssgsea was used to estimate the relative expression of 18 different TME cell types by using the Consensus^TME^ [23,24], which integrates seven different methods of gene sets or TME cell type estimation methods.

### 4.6. Outcomes Classification through Machine Learning Classifiers

#### 4.6.1. Combined Intra-Tumor and Inter-Tumor Site Radiomic, Clinical, and Genomic (iRCG) Score of PFS

A generalized linear model (GLMNet) [49] using elastic net feature selection constraints was fit using cluDiss, conventional clinical, and genomic variables to classify patient survival in months using MSKCC as training dataset. Best tuning parameters ∝ and λ (or the L1-norm penalty) were determined from the training set (MSKCC). The relative feature importance obtained from this model was used to combine cluDiss, clinical, and genomic variables into a single continuous iRCG_PFS_ score. An optimal cut point was determined from the iRCG score using receiver operating characteristic curve (ROC) analysis (optimalCutpoints in R) on the MSKCC set and applied on the external TCIA dataset to dichotomize patients into low-risk and high-risk groups. The same approach was repeated by combining conventional imaging and average heterogeneity radiomic measures with clinical and genomic variables to extract CCG_PFS_ and aRCG_PFS_ scores, respectively.

#### 4.6.2. Platinum Resistance Classification

Sixty-one of the 75 patients had platinum resistance information (Table 1) with 14 patients resistant while the remaining 47 are sensitive to platinum resistance chemotherapy. Since the number of patients for training a machine learning classifier were relatively small, and due to the large class imbalance, this analysis was performed using cross-validation instead of using the TCIA dataset as a hold-out testing set, as done for determining PFS association in Section 4.6.1. This approach is valid for the purpose of this work where the goal was to assess the utility of the cluDis measure in comparison to conventional clinical and radiomics measures.

iRCG-SVM: A linear support vector machine classifier (SVM) [50] was constructed by combining cluDiss with three clinical (age, stage, cytoreductive outcome), and genomic (CNB) variables. A linear SVM was used since it treats the individual factors independently and avoids overfitting in small datasets. The classifier was trained with three-fold cross validation with 100 repetitions. Class imbalance was handled by the synthetic minority oversampling (SMOTE) technique as used in our prior work for classifying prostate cancer aggressiveness using radiomics measures [17].

Conventional-Clinical-Genomic SVM (CCG-SVM): A conventional-clinical-genomic linear SVM classifier of platinum resistance was trained using the clinical, conventional imaging, and CNB variables.

aRCG-SVM: A recursive feature elimination linear SVM classifier (RFE-SVM) was trained with repeated (100 repetitions) and nested (three-fold outer and three-fold inner) cross validation using average heterogeneity radiomic features found to be robust to scanner differences (Table 1), clinical, and CNB variables. Nested cross-validation was done to select the appropriate number of features (*N* = 5, 10, 15, 20, 25). The SMOTE method was used to handle class imbalance. RFE-SVM was used to perform implicit feature selection from the relatively large number of features used within the classifier.

### 4.7. Feature Robustness to CT Manufacturer

Statistical differences to CT scanners (GE vs. non-GE) in cluDiss and average heterogeneity radiomic features were evaluated. All MSKCC patients and 24 out of 38 TCIA patients were scanned on GE. The remaining 12 were scanned on Siemens and one each on Toshiba and Philips scanners. The goal of this experiment was to evaluate whether the variability in the features was due to the different CT scanner manufacturers. This is because large feature variabilities between scanners can obscure the signal to differentiate between the outcomes, thereby reducing the performance of radiomics measures [51].

### 4.8. Statistical Analysis

Machine learning classifiers were trained with repeated three-fold cross-validation and nested cross-validation (where applicable) to reduce bias in classification. Accuracy was computed from samples not used in training in each fold. Area under the receiver operating characteristic curve (AUROC), sensitivity, and specificity with 95th percentile confidence intervals were computed. DeLong’s method [52] was used to measure the differences in classifiers’ AUC. Patient characteristics and texture measures were summarized using median and interquartile range (IQR). Data with missing variables or outcomes were excluded from the analysis. Two-sided Wilcoxon rank-sum test was used to test radiomics differences to scanners. Only *p* > 0.05 were considered significant.

Cox proportional hazard regression analysis was performed to test association with PFS using the iRCG, CCG, and aRCG scores. Hazard ratios (HR) and 95% confidence intervals were estimated. Dichotomized groups generated according to the cut points were used to compute Kaplan-Meier curves on the TCIA dataset. *p* Values < 0.05 were considered statistically significant. Concordance probability estimates (CPE) were computed for the individual predictors for determining the strength of association with survival measures.

Association between radiomic and gene set pathways were computed using Spearman rank correlation coefficients and principal component analysis of the Hallmark gene sets was performed using *factoMineR* software in R after scaling of the gene expressions.

All statistical analyses were performed in the software packages R, version 3.4 (The R Foundation for statistical computing). The code for textures and IISTH computation is available through open source software CERR [46]. 

## 5. Conclusions

We validated a previously developed intra-site and inter-site tumor heterogeneity measure (cluDiss) for predicting HGSOC outcomes. We showed that cluDiss, combined with known clinical and genomic variables, outperformed conventional clinical-genomic and standard radiomic-clinical-genomic models in predicting HGSOC outcomes. This measure was negatively correlated to *Wnt* signaling and positively to immune TME cell types. Validation on larger, multi-institutional cohorts is necessary to verify the potential for patient stratification.

## Figures and Tables

**Figure 1 cancers-12-03403-f001:**
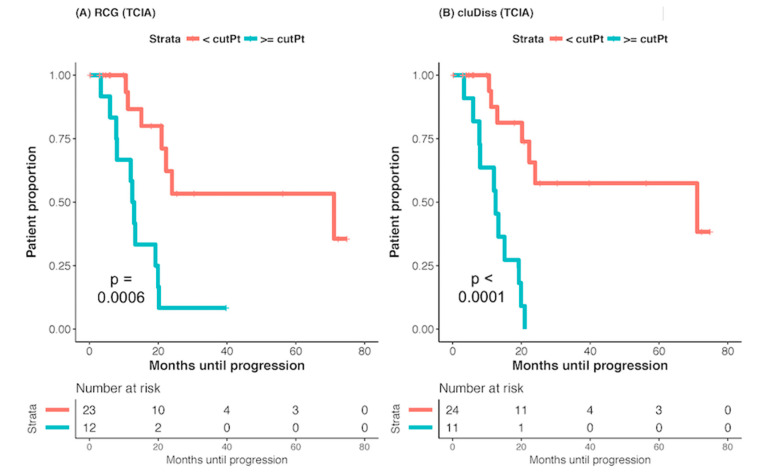
Kaplan-Meier curves computed using (**A**) dichotomized iRCG_PFS_ cut point (low risk < 642) and (**B**) cluDiss cut point (low risk < 68.82). The cut points were determined on the MSKCC dataset and applied to the TCIA dataset.

**Figure 2 cancers-12-03403-f002:**
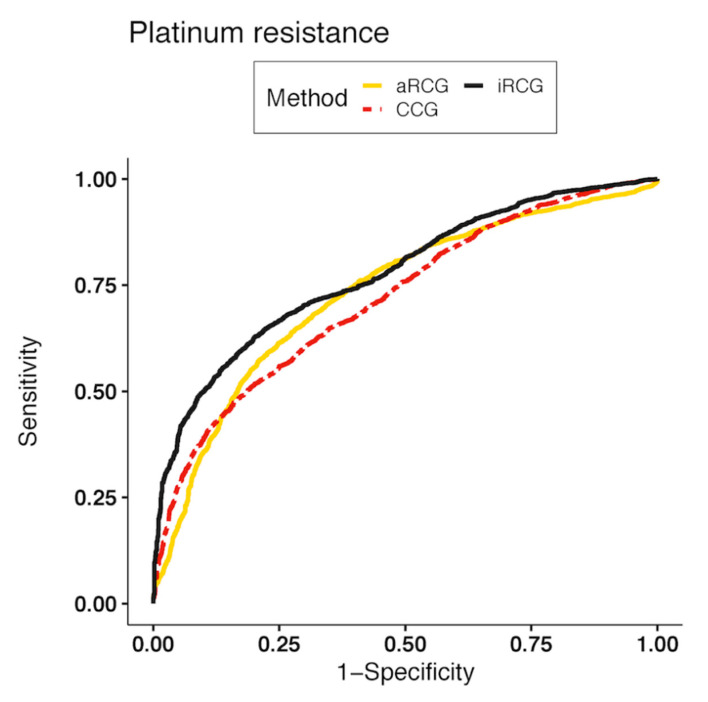
Receiver operating characteristic (ROC) curves for classifying patients by platinum resistance using the iRCG-SVM, CCG-SVM, and aCCG-SVM classifiers.

**Figure 3 cancers-12-03403-f003:**
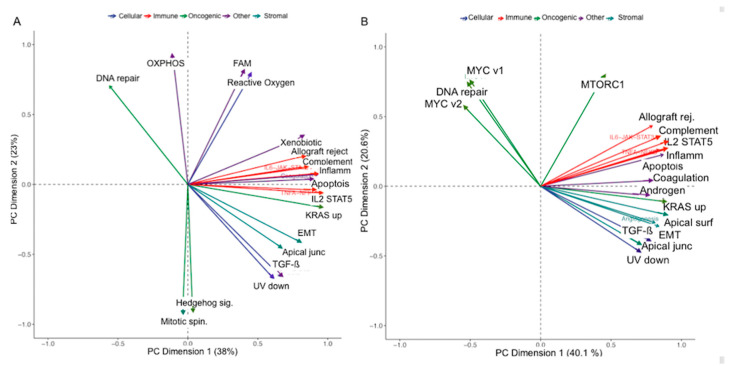
Principal component analysis (PCA) loadings of the Hallmark gene sets on (**A**) low-(cluDiss < 68.6) and (**B**) high-risk (cluDiss ≥ 68.6) patient groups. Only the top 20 contributing gene set pathways are shown.

**Figure 4 cancers-12-03403-f004:**
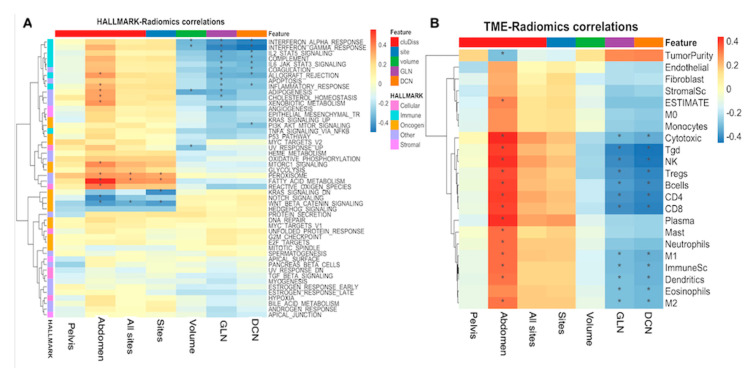
Spearman rank correlation coefficient matrix of the relevant radiomic measures and (**A**) 50 HALLMARK gene sets and (**B**) the consensus^TME^ TME cell types. Significant correlations (*p* < 0.05) are indicated with *.

**Figure 5 cancers-12-03403-f005:**
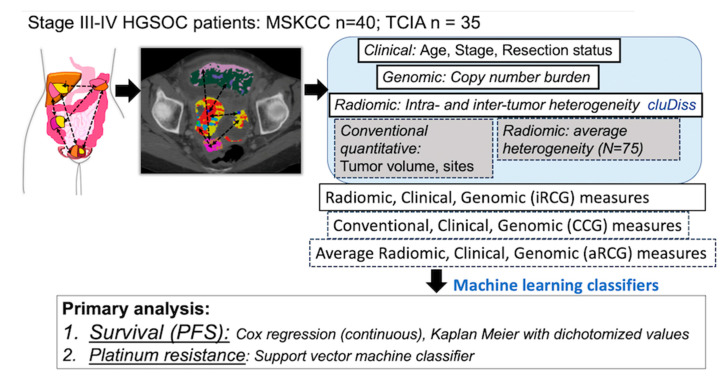
Schema of experimental workflow. The intra-tumor and inter-tumor radiomic heterogeneity (IISH) measures that summarize the heterogeneity across the various sites of disease are combined with clinical and genomic factors to produce a combined classifier of outcome.

**Table 1 cancers-12-03403-t001:** Patient characteristics of 75 analyzed patients.

Patient Characteristics	MSKCC (*N* = 40)	TCIA (*N* = 35)
Age (median) (IQR)	59 (50.8–66)	61 (52–71)
Stage at diagnosis (proportion patients)		
III	27 (67.5%)	31 (88.6%)
IV	13 (32.5%)	4 (11.4%)
Surgical debulking outcome (number of patients)		
Complete	14	8
Optimal	20	16
Suboptimal	6	11
Recurrence status * (number of patients)		
Recurring	38	18
Not recurring	2	17
Disease status (number of patients)		
Alive	17	14
Dead	23	21
Follow up * mos (median) (IQR)	41.9 (22.9–56.3)	19.3 (6.3–38.6)
Survival (median) (IQR)		
PFS ^+^ mos	15.4 (10.5–26.2)	13.3 (7.0–21.6)
OS ^+^ mos	59.0 (43.1–76.4)	30.0 (14.5–53.1)
Platinum status (number of patients)		
Sensitive	31	16
Resistant	7	7
Unknown	2 ^§^	12 ^§^
Tumor volume (cm^3^) * (median) (IQR)	122.0 (65.5–229.0)	331.0 (158.2–595.0)
Tumor sites (median) * (IQR)	7 (6–9)	4 (3–5)
Copy number alterations (median) (IQR)	0.546 (0.446–0.653)	0.584 (0.443–0.654)
CT scanners: GE	40	21
Siemens Philips or Toshiba	0 0	12 2

* indicates datasets were significantly different (*p* < 0.05), + indicates datasets were significantly different (*p* < 0.05) computed using Log-rank tests. The reported number of events occurring within the time frame of the study. ^§^ These cases were removed for platinum resistance classification, and 61 remaining cases were used. Abbreviations: IQR – Inter quartile range; PFS – progression free survival; OS – overall survival.

**Table 2 cancers-12-03403-t002:** Univariate and multivariable associations of computed radiomic measures with progression-free survival (PFS).

Variable	Univariate Analysis				Multivariable Analysis			
	MSKCC		TCIA		MSKCC		TCIA	
	*p*-Value	HR (95% CI)	*p*-Value	HR (95% CI)	*p*-Value	HR (95% CI)	*p*-Value	HR (95% CI)
cluDiss	0.0025	1.02 (1.01, 1.03)	0.002	1.03 (1.01, 1.05)	0.0008	1.03 (1.01, 1.04)	0.004	1.04 (1.01, 1.07)
Number of sites	0.049	1.13 (1.00, 1.28)	0.029	1.59 (1.05, 2.40)	0.242	1.11 (0.94, 1.31)	0.009	2.00 (1.19, 3.37)
TTV	0.705	1.06 (0.78, 1.44)	0.653	0.914 (0.62, 1.35)	0.813	0.953 (0.64, 1.42)	0.513	0.855 (0.54, 1.37)
iRCG	0.0004	1.36 (1.15, 1.61)	0.007	1.39 (1.10, 1.76)	0.001	1.38 (1.13, 1.68)	0.009	1.46 (1.10, 1.93)
CCG	0.058	1.34 (0.9, 1.81)	0.515	1.13 (0.77, 1.66)	0.411	0.97 (0.91, 1.04)	0.825	1.02 (0.87, 1.19)
aRCG	0.478	0.98 (0.92, 1.04)	0.863	0.99 (0.86, 1.13)	0.539	0.82 (0.44, 1.53)	0.68	1.16 (0.56, 2.40)

Abbreviations: CI–confidence interval. HR–hazard ratio. TTV–Total tumor volume. iRCG–intra-inter radiomic, conventional clinical, genomic classifier. CCG–conventional clinical, genomic classifier. aRCG–average heterogeneity radiomic, conventional clinical, and genomic classifier.

**Table 3 cancers-12-03403-t003:** Machine learning classifier accuracies using intra-inter site radiomic-clinical-genomic (iRCG), conventional-clinical-genomic (CCG), and average radiomic-clinical-genomic (aRCG) classifiers of platinum resistance.

Method	AUROC (95% CI)	Sensitivity (95% CI)	Specificity (95% CI)	*p*-Value (Method vs. iRCG)
iRCG SVM	0.78 (0.76, 0.79)	0.75 (0.72, 0.77)	0.66 (0.65, 0.68)	
CCG SVM	0.72 (0.70, 0.73)	0.66 (0.64, 0.69)	0.65 (0.64, 0.67)	<0.001
aRCG SVM *	0.73 (0.72, 0.75)	0.68 (0.66, 0.71)	0.62 (0.60, 0.63)	<0.001

* Recursive feature elimination support vector machine (SVM) classifier was used due to a large number of features to perform implicit feature selection.

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
