# Peer review of "Integrated Multi-Tumor Radio-Genomic Marker of Outcomes in Patients with High Serous Ovarian Carcinoma"

_cancers, 2020, doi:10.3390/cancers12113403_

Round 1
Reviewer 1 Report
This is an interesting preliminary study which has developed a model, CluDiss and iRCG, which is suggested to provide better prognostication of HGSOC than conventional and average radiomic measures.
Overall, the manuscript is written well and the results are clearly presented. Although, the sample size is small, this limitation is noted by the authors, the pilot study provides sufficient data to facilitate further studies in this direction.
Few comments for the authors are noted below:
1. Could the authors further clarity if and how the 50 gene set, extracted using ssgsea, could dissect the inter- or intra- tumor heterogeneity?
2. Did the authors stain any tumor samples for infiltrating immune cells?
3. Were the results validated using IHC or in-situ assays such as RNAscope related to the 50 gene set?
4. It would be valuable to add to the discussion that, How did the authors define heterogeneity in their study and in the samples used for the study? what were the criteria? and based on that how heterogeneous were the samples? this would help in further understanding the applicability of the method suggested in this study.
5. lines 238-242: Can the authors comment which immune cells types were positively correlated with cluDiss? Was there a negative correlation seen with any immune cell type?
Author Response
We thank the associate editor and the reviewers for the comments and suggestions. It is encouraging that the reviewers found the manuscript relevant and interesting.
We have responded to all of the reviewers’ comments and suggestions to the best of our ability. The changes made in lieu of the reviewer suggestions are indicated with blue font in the paper.
Reviewer 1
This is an interesting preliminary study which has developed a model, CluDiss and iRCG, which is suggested to provide better prognostication of HGSOC than conventional and average radiomic measures. Overall, the manuscript is written well and the results are clearly presented. Although, the sample size is small, this limitation is noted by the authors, the pilot study provides sufficient data to facilitate further studies in this direction.
Thank you for the encouraging comments.
Q1. Could the authors further clarity if and how the 50 gene set, extracted using ssgsea, could dissect the inter- or intra- tumor heterogeneity?
These gene sets are candidates for genes that may either drive genomic heterogeneity or are required for survival in the context of ongoing chromosomal instability. The genomic samples were collected only from a single primary tumor site using common guidelines in The Cancer Genome Atlas (TCGA) study for ovarian cancers (“Integrated genomic analyses of ovarian carcinoma”, The cancer genome atlas network, Nature 2011). As described in the TCGA study, biospecimens from collected from patients newly diagnosed with ovarian serous adenocarcinoma and undergoing surgery. One tumor specimen and matched normal tissue specimen was collected for each patient.
Dissecting the intra- tumor heterogeneity would require multiple samples from distinct locations in the patient, while quantifying the inter-tumor heterogeneity would require samples from both primary and metastatic tumor sites in the patient. Both of these are difficult to perform other than in a small number of patients as previously reported in the works of Bashashati et.al J Pathol 2013, Schwarz et.al Plos Medicine 2015, Jimenez-Sanchez et.al Cell 2018, and Jimenez-Sanchez et.al Nature Genetics 2020.
The difficulty in performing multi-tumor genomic studies on a larger scale is the motivation for developing the image-based biomarker that captures the imaging heterogeneity both within the spatially and texturally distinct regions within and between tumor sites in the patient. We have clarified our discussion in the paper as follows:
“Quantifying inter and intra-tumor imaging heterogeneity is important because HGSOC exhibits widespread genomic intra-tumoral heterogeneity. Multi-site genomic studies have shown intratumor genomic heterogeneity to correlate to poor survival [7]. In addition to clonal heterogeneity, the malignant cell spread within the peritoneal cavity is distinct and non-random[8, 9], as some sites harbor genetically diverse clones[8]. These site-specific properties, including immunologic components of the TME, may modulate malignant cell invasion and expansion, thereby shaping evolutionary selection[28]. However, large scale multi-site genomic heterogeneity studies are difficult to do and are impractical for clinical practice. This motivated the development and validation of a non-invasive CT based measure of intra- and inter-site heterogeneity.” and
“Understanding the radiogenomic correlations are important for furthering their application as biomarkers of treatment response[11]. Multiple studies have reported the association of image-based qualitative[31] and quantitative radiomics features with genomic measures [32-36]. We measured the correlation of cluDiss and two radiomics measures DCN and GLN with the Hallmark gene sets and TME cell types extracted from ssgsea analysis of the RNA expressions. Gene sets are candidates for genes that may either drive genomic heterogeneity or are required for survival in the context of ongoing chromosomal instability. Patients dichotomized into low and high-risk groups using cluDiss showed a difference in the gene set pathways. Also, cluDiss was negatively correlated with Wnt signaling pathway and positively to 13 immune TME cell types including Tregs, Tgd, Bcells, CD4, CD8, and NK. Prior work by our group[25] showed a mutual exclusivity in the expression of Wnt and Myc gene pathways with respect to the immune cell types in untreated HGSOC patients. CluDiss has also been shown to correlate with CKB protein in a previous study using matched imaging and proteomic samples from 20 HGSOC patients by our group [37].”
Q2. Did the authors stain any tumor samples for infiltrating immune cells?
No, we did not perform any staining as all the mRNA expressions were available from the open-source The Cancer Genomics Archive (TCGA).
Q3. Were the results validated using IHC or in-situ assays such as RNAscope related to the 50 gene set?
No, we did not perform validation of the 50 gene sets to the IHC due to limited number of samples.
Q4. It would be valuable to add to the discussion that, How did the authors define heterogeneity in their study and in the samples used for the study? what were the criteria? and based on that how heterogeneous were the samples? this would help in further understanding the applicability of the method suggested in this study.
The genomic samples themselves were collected prior to even conceptualizing this study by the Cancer Genome Atlas group (TCGA) and only from the primary tumor site. Hence, we did not set any specific criteria on the genomic heterogeneity and used only the patients who had matched CT imaging and genomic analysis done for performing the radiogenomic correlation analysis.
Our goal in this study was to evaluate whether the imaging heterogeneity quantified from the primary and metastatic sites is associated with cancer outcomes. In a subset of patients with matched CT imaging and genomics, we explored the relation of imaging heterogeneity to the gene sets extracted from the mRNA expressions. The imaging heterogeneity was defined using the magnitude of the cluDiss variable, where larger values indicate larger imaging textural differences between the tumor sites and within the tumor sites. We have clarified this in the discussion of the paper.
“Unlike most radiomic studies[10, 12, 16] that compute an average measure of single tumor heterogeneity, cluDiss quantifies the heterogeneity within and between the entire tumor burden rather than just the ovarian mass. It’s magnitude increases with the number of sites and the textural differences between the sub-regions within and between the tumor sites, reflecting larger imaging heterogeneity.”
Q5. lines 238-242: Can the authors comment which immune cells types were positively correlated with cluDiss? Was there a negative correlation seen with any immune cell type?
We included the results of measuring the correlation of cluDiss and other relevant radiomics measures to 18 different TME cell types computed using the consensusTME method that was developed by Jiménez-Sánchez et.al, Cancer Research 2019, “Comprehensive benchmarking and integration of tumor microenvironment cell estimation methods”. The following was added to the results:
“CluDiss measure from the pelvic sites showed no significant correlation to any of the gene set pathways. CluDiss measure computed from the abdominal metastases was negatively correlated with Wnt and NOTCH signaling (Figure 4, Supplementary Table S3), positively correlated with MTORC1 and allograft rejection (Figure 4), immune cells (ρ= 0.33, p = 0.028). This same measure was positively correlated to 13 of the 18 TME cell types (Figure 4, Supplementary Table S4), including Tgd (ρ= 0.40, p = 0.007), Treg (ρ= 0.39, p = 0.009), Bcells (ρ= 0.39, p = 0.008), CD4 (ρ= 0.41, p = 0.006), CD8(ρ= 0.40, p = 0.007), cytotoxic (ρ= 0.44, p = 0.003), and NK(ρ= 0.43, p = 0.004) cell types. DCN and GLN both of which measure the heterogeneity in the distribution of the local signal intensities are highly correlated (ρ= 0.98, p < 0.0001) with each other. DCN was negatively correlated with immune gene-sets (Figure 4) (Supplementary Table S3), immune cells (ρ= -0.32, p = 0.033), and 11 of the TME cell types (Figure 4, Supplementary Table S4), including Tgd (ρ= -0.46, p = 0.002), Treg (ρ= -0.42, p = 0.005), Bcells (ρ= -0.41, p = 0.006), CD4 (ρ= -0.41, p = 0.005), CD8(ρ= -0.42, p = 0.004), cytotoxic (ρ= -0.34, p = 0.02), and NK(ρ= -0.45, p = 0.002) cell types.”
Reviewer 2 Report
This manuscript describes the validation of a previously developed intra-inter-tumor radiomics measure (cluDiss) in 75 HGSOC patients from two independent cohorts. The authors further integrated cluDiss with three clinical variables (age, stage, cytoreductive outcome) and tumor copy number burden to develop the integrated radio-clinical-genomic (iRCG) measure. The novel measures were then compared against conventional and average radiomics models to evaluate their goodness in prognosticating progression free survival (PFS) and platinum resistance. Finally, the authors searched the expressional differences in the molecular signaling pathways between patients with low or high risk of rapid disease progression.
Broad comments:
The imaging of the inter- and intratumor heterogeneity is highly relevant for HGSOC and the integration of radiomic, clinical and genomic information for prognostication of HGSOC is interesting. Although the results actually show that the genomic information (copy number burden) has none or very limited value in the computing models, the combination of radiomic (cluDiss) and clinical data may form a non-invasive way to prognosticate disease progression. However, the genomic information may be used to find pathways that are activated or suppressed in the patient who are classified to be in high risk of rapid progression using the abovementioned models.
Although the study is well designed and presented there are some concerns to be clarified. Specifically, there is a need to reformulate some interpretations of the results.
Broad comments:
- Please discuss the benefit reached by adding the clinical and genomic data (iRCG) to cluDiss. For my eyes there is no benefit for KM-curve for progression, little for concordance probability estimate, clear benefit for univariate model but unknown in multivariate model). It was concluded that CNB is not relevant for the models, but this was not further discussed. It is also of importance if neither clinical nor genomic variables bring significant additive information on prediction of PFS or platinum resistance vs. cluDiss alone.
- I could not find the information whether the copy number burden (CNB) was available for one or multiple tumors of each patient? How much CNB vary between tumors within a patient? From that perspective, would the unified tumor location give stronger effect for CNB in the models? Similarly, how about the variability of tumor locations in the gene enrichment analysis?
Specific comments:
- Abstract: When HR is only slightly above 1 (1.03) I would not call the association with PFS to be strong although it is clearly significant. Please modify.
- Table 1: Would it be more relevant to present the table with the 75 patients included in all the analyses? If all the 83 patients are used in any analysis and therefore rationale the current table 1, please clarify the patient number used in each analysis.
- Table 2 and Line 144 onwards: iRCG is missing from the multivariate analysis although it has highest and significant hazard ratio in the univariate analysis. On the other hand, TTV is included in the multivariate analysis although it is not significant even in univariate analysis. Is it a mistake?
- Line 147: “The number of sites showed association with PFS in both datasets in the univariate analysis but not in the multivariate analysis.” However, for TCIA cohort the HR = 2 and p=0.002 in the multivariate analysis. Please correct.
- Line 155: Please check the first sentence. Also, when the 95% CI of the CPE for cluDiss and number of sites is fully overlapping (as for TCIA cohort), it should not be considered different (“higher”).
- Figure 2: What is the rationale for using smooth lines in the ROC curve? Please explain or prefer commonly used straight lines, which gives more detailed information about the specificity and sensitivity.
- Line 189: “CNB was not relevant in all three models.” All or any of the models?
- Line 193: I would not call cluDiss specifically robust for scanner differences if p is close to be significant (p=0.06). I suggest modifying the sentence. I was also a bit worried using all the 75 radiomic measures of cluDiss in iRCG-SVM and only the 24 robust measures for aRCG-SVM when you compare the two models. However, as long as the aRCG-SVM is clearly weaker in finding PFS than iRCG-SVM, it should not significantly change the result and conclusions of the study. Am I correct?
- Line 203: Were the risk groups assessed by cluDiss or iRCG-PFS?
- I see a discrepancy here: Results Line 188-189: “All variables except CNB were relevant in the iRCG model” (in classification of platinum resistance) and “CNB was not relevant in all three models.” vs. Discussion line 264: More importantly, our results show that combining multi-tumor radiomic (using cluDiss), clinical, and genomic variables more accurately predicted HGSOC outcomes than conventional imaging (total tumor volume and number of sites) and average tumor heterogeneity radiomics models. Please modify the discussion to be in line with the results.
Author Response
Reviewer 2
This manuscript describes the validation of a previously developed intra-inter-tumor radiomics measure (cluDiss) in 75 HGSOC patients from two independent cohorts. The authors further integrated cluDiss with three clinical variables (age, stage, cytoreductive outcome) and tumor copy number burden to develop the integrated radio-clinical-genomic (iRCG) measure. The novel measures were then compared against conventional and average radiomics models to evaluate their goodness in prognosticating progression free survival (PFS) and platinum resistance. Finally, the authors searched the expressional differences in the molecular signaling pathways between patients with low or high risk of rapid disease progression.
Broad comments:
Q1. The imaging of the inter- and intratumor heterogeneity is highly relevant for HGSOC and the integration of radiomic, clinical and genomic information for prognostication of HGSOC is interesting. Although the results actually show that the genomic information (copy number burden) has none or very limited value in the computing models, the combination of radiomic (cluDiss) and clinical data may form a non-invasive way to prognosticate disease progression. However, the genomic information may be used to find pathways that are activated or suppressed in the patient who are classified to be in high risk of rapid progression using the above mentioned models.
The aim of our analysis was not to downplay the genomic information (CNB) but to evaluate whether its combination with the radiomics features could improve the accuracy of classification of PFS and platinum resistance more than cluDiss alone. The copy number burden (CNB) was used in the combined iRCG model for classifying progression free survival as we showed in Section 2.2. However, it was not found to be relevant for the iRCG (intra- and inter-site radiomics-clinical-genomics) and the CCG (conventional-clinical-genomics) model for classifying platinum resistance and had the lowest importance in the aRCG (average radiomics-clinical-genomics) model. But, our dataset is still limited and we plan to further validate this in future studies.
We have now clarified this in the discussion section of the paper.
“Although cluDiss was as good as the iRCGPFS measure for predicting PFS, the platinum resistance classification benefitted from the clinical measures, indicated by their higher importance over cluDiss for that model. On the other hand, CNB, while relevant for the iRCGPFS model, was not relevant for classifying platinum resistance. Both cluDiss and the clinical measures can be obtained in a non-invasive way and their combination could thus serve to non-invasively stratify patient outcomes without needing genomic measures. On the other hand, genomic measures could be used for obtaining a mechanistic drivers of risk in those patients determined to be high risk using the non-invasive measures, such as by finding activated or suppressed pathways.”
Q2. Although the study is well designed and presented there are some concerns to be clarified. Specifically, there is a need to reformulate some interpretations of the results.
We thank the reviewer for the suggestions. We have made changes in the interpretation and presentation of results as suggested by the reviewer.
Broad comments:
Q3. Please discuss the benefit reached by adding the clinical and genomic data (iRCG) to cluDiss. For my eyes there is no benefit for KM-curve for progression, little for concordance probability estimate, clear benefit for univariate model but unknown in multivariate model). It was concluded that CNB is not relevant for the models, but this was not further discussed. It is also of importance if neither clinical nor genomic variables bring significant additive information on prediction of PFS or platinum resistance vs. cluDiss alone.
CNB was not found to be important in the classifiers of platinum resistance. But it was relevant to the model used for PFS classification. We have clarified this in the manuscript. While there was little difference between the iRCG and cludiss for progression free survival, the clinical variables, particularly the resection status was the most important for predicting platinum resistance. We have clarified this in the both results and discussion in the manuscript.
In results:
“The iRCGPFS produced the highest CPE of all other integrated models. It was slightly better than the cluDiss measure alone.”
“All variables except CNB were relevant (importance > 0) in the iRCG and CCG models. Resection status was the most relevant feature for all three models (Importance = 100), while cluDiss had a lower importance of 34.4, clearly indicating the relevance of the clinical variables for predicting platinum resistance. Two radiomic measures, dependence counts non-uniformity (DCN) and the gray level non-uniformity (GLN) were found to be relevant in the aRCG model. CNB had a low importance of 2.32 in the aRCG model.”
In discussion:
“Although cluDiss was as good as the iRCGPFS measure for predicting PFS, the platinum resistance classification benefitted from the clinical measures, indicated by their higher importance over cluDiss for that model.”
Q4. I could not find the information whether the copy number burden (CNB) was available for one or multiple tumors of each patient? How much CNB vary between tumors within a patient? From that perspective, would the unified tumor location give a stronger effect for CNB in the models? Similarly, how about the variability of tumor locations in the gene enrichment analysis?
The genomic samples were collected only from a single primary tumor site using common guidelines in The Cancer Genome Atlas (TCGA) study for ovarian cancers (“Integrated genomic analyses of ovarian carcinoma”, The cancer genome atlas network, Nature 2011). As described in the TCGA study, biospecimens from collected from patients newly diagnosed with ovarian serous adenocarcinoma and undergoing surgery. One tumor specimen and matched normal tissue specimen was collected for each patient. The questions the reviewer raised regarding the impact of tumor locations on CNB and the variability of tumor locations on gene set enrichment are all very interesting. However, as all of the analysis was performed with retrospectively collected samples available through TCGA-TCIA, it was not possible for us to perform any study of variability of CNB or gene set pathways across the tumor sites.
We have clarified this in the manuscript materials and methods
“CNB was computed from the genomic sample taken from a single primary tumor site using common guidelines in the TCGA ovarian cancer study[41]. Specifically, biospecimens were collected from newly diagnosed ovarian cancer serous adenocarcinoma patients undergoing surgery. One tumor and matched normal tissue specimen were collected for each patient.”
and in discussion:
“Also the genomic samples were available from only a single primary tumor site. Hence, a study of variability in gene sets and CNB between tumor sites, or their impact on classifying outcomes was not possible.”
Specific comments:
Q5. Abstract: When HR is only slightly above 1 (1.03) I would not call the association with PFS to be strong although it is clearly significant. Please modify.
Done. We changed this to “CluDiss was associated with PFS”
Q6. Table 1: Would it be more relevant to present the table with the 75 patients included in all the analyses? If all the 83 patients are used in any analysis and therefore rationale the current table 1, please clarify the patient number used in each analysis.
We have updated the table with 75 patients. We have also clarified the number of patients used in each analysis in the table in addition to that reported in the individual experiments before.
“After excluding 14 patients who had no platinum resistance data, 61 patients were analyzed for platinum resistance classification. Forty four cases had matched imaging and RNA-sequencing data and used for radiogenomic analysis.”
Q7. Table 2 and Line 144 onwards: iRCG is missing from the multivariate analysis although it has highest and significant hazard ratio in the univariate analysis. On the other hand, TTV is included in the multivariate analysis although it is not significant even in univariate analysis. Is it a mistake?
We have updated the table with the multivariable analysis for the iRCG, aRCG, and CCG measures.
Q8. Line 147: “The number of sites showed association with PFS in both datasets in the univariate analysis but not in the multivariate analysis.” However, for TCIA cohort the HR = 2 and p=0.002 in the multivariate analysis. Please correct.
Thank you for pointing this out. We have now corrected this.
Q9. Line 155: Please check the first sentence. Also, when the 95% CI of the CPE for cluDiss and number of sites is fully overlapping (as for TCIA cohort), it should not be considered different (“higher”).
Done. Changed the sentence to:
“The cluDiss measure achieved a concordance probability estimate (CPE) for PFS of 0.66 (95% CI of 0.61 to 0.70) for MSKCC and 0.67 (95% CI of 0.63 to 0.72) for TCIA cohorts. The CPE for number of sites was (MSKCC CPE of 0.59 [95% CI of 0.55 to 0.64]; TCIA CPE of 0.655 [95% CI of 0.54 to 0.77]) and TTV was (MSKCC CPE of 0.52 [95% CI of 0.47 to 0.57]; TCIA CPE of 0.535 [95% CI of 0.46 to 0.61]), respectively.”
Q10. Figure 2: What is the rationale for using smooth lines in the ROC curve? Please explain or prefer commonly used straight lines, which gives more detailed information about the specificity and sensitivity.
We have replaced Figure 2 with the straight line ROC curve.
Q11. Line 189: “CNB was not relevant in all three models.” All or any of the models?
We have clarified the text in the manuscript. The text is now modified as:
“All variables except CNB were relevant (importance > 0) in the iRCG and CCG models. Resection status was the most relevant feature for all three models (Importance = 100), while cluDiss had a lower importance of 34.4, clearly indicating the relevance of the clinical variables for predicting platinum resistance. Two radiomic measures, dependence counts non-uniformity (DCN) and the gray level non-uniformity (GLN) were found to be relevant in the aRCG model. CNB had a low importance of 2.32 in the aRCG model.”
Q12. Line 193: I would not call cluDiss specifically robust for scanner differences if p is close to be significant (p=0.06). I suggest modifying the sentence. I was also a bit worried using all the 75 radiomic measures of cluDiss in iRCG-SVM and only the 24 robust measures for aRCG-SVM when you compare the two models. However, as long as the aRCG-SVM is clearly weaker in finding PFS than iRCG-SVM, it should not significantly change the result and conclusions of the study. Am I correct?
We apologize for any confusion. We have edited the text to state that cluDiss did not show statistical difference between scanners in the text.
The cluDiss measure was not combined with the 75 radiomics measures in the iRCG-SVM. The 75 standard radiomic measures were evaluated separately in the cRCG model. The iRCG-SVM had far fewer features (only cluDiss, clinical measures age, stage, resection, and CNB) than the aRCG-SVM model, which had 24 radiomic measures, the clinical measures age, stage, resection, and the CNB.
Q13. Line 203: Were the risk groups assessed by cluDiss or iRCG-PFS?
They were assessed by cluDiss.
Q14. I see a discrepancy here: Results Line 188-189: “All variables except CNB were relevant in the iRCG model” (in classification of platinum resistance) and “CNB was not relevant in all three models.” vs. Discussion line 264: More importantly, our results show that combining multi-tumor radiomic (using cluDiss), clinical, and genomic variables more accurately predicted HGSOC outcomes than conventional imaging (total tumor volume and number of sites) and average tumor heterogeneity radiomics models. Please modify the discussion to be in line with the results.
Thank you for pointing this out. We have updated and clarified this in the manuscript.
“More importantly, our results show that the integrated model combining cluDiss, clinical, and genomic variables was more accurate than the conventional imaging (total tumor volume and number of sites) and average tumor heterogeneity radiomics models for predicting PFS.”
Reviewer 3 Report
Title: The meaning of “multi-tumor” is unclear to me in this setting. Can the authors explain why they want to use this term here? Is there any other available term?
Simple Summary: It seems that the developed ML tool integrating CT radiomics, clinical and genomic data is more accurate than radiomic metrics alone, which is not surprising. Maybe authors should state that their integrated approach is more accurate than standard-of-care estimates progression-free survival and platinum resistance (clinical and average radiomics).
Abstract:
- The term “inter-tumor” might be confusing. Can the authors explain why is it use here? Does it refer to “inter-site”?
- Again, the term “intra-inter-tumor” confuse me here. Can it be clarified?
- Do authors expect and improvement in patients management base on the higher accuracy of their iRCG model?
Introduction:
100-102. Authors should clarify if inter-, intra- intra-and-inter-tumor is equal to -site. It might be easier to use site as representative of primary and secondary tumours.
Results:
- At the abstract authors stated that there were 75 cases but here it is stated that there were 83 patients and only 44 were used for the radiogenomic analysis and machine learning. This must be clarified if relevant for the iRCG marker. Patients should also be matched with Table 1.
- The number of images seems not to be relevant. I guess there was 1 CT exam per patient.
- The expected improvements in patients’ management by using the iRCG marker should be described.
- How was the robustness to scanner differences calculated if test-retest exams were not performed? This must be clarified (also, 448).
Discussion:
Regarding bias, as some radiomic features might vary with different CT protocols, protocols have to be commented (mean values and ranges, contrast doses, acquisition delays).
Materials and Methods:
- 75 patients here. Please, clarify.
Conclusions: OK
References: OK
Figures and Tables: OK
Author Response
Reviewer 3
Q1. Title: The meaning of “multi-tumor” is unclear to me in this setting. Can the authors explain why they want to use this term here? Is there any other available term?
The concept of multi-tumor or multi-site is a very important one, especially in HGSOC where patients present with multi-site disease which is heterogeneous both within the site and between the sites of disease. There is now evidence that this profound both intra- and inter-site heterogeneity drives resistance to chemotherapy. The cluDiss is a measure of textural heterogeneity within and between tumor sites in each patient. We have clarified this in the manuscript and throughout used the term intra- and inter-site heterogeneity (IISH).
Q2. Simple Summary: It seems that the developed ML tool integrating CT radiomics, clinical and genomic data is more accurate than radiomic metrics alone, which is not surprising. Maybe authors should state that their integrated approach is more accurate than standard-of-care estimates progression-free survival and platinum resistance (clinical and average radiomics).
Thank you for the suggestion. We have updated this in the manuscript.
Abstract:
Q3. The term “inter-tumor” might be confusing. Can the authors explain why is it use here? Does it refer to “inter-site”?
We apologize for the confusion. The cluDiss is a measure of textural heterogeneity within and between tumor sites in each patient. We have clarified this in the manuscript and refer to this uniformly as intra- and inter-site radiomic heterogeneity. We have modified the sentence as:
“To develop an integrated intra and inter-site radiomic-clinical-genomic (iRCG) marker of high grade serous ovarian cancer (HGSOC) outcomes and explore the biological basis of radiomics with respect to molecular signaling pathways and the tumor microenvironment (TME).”
Q4. Again, the term “intra-inter-tumor” confuse me here. Can it be clarified?
We have fixed this to intra- and inter-site radiomics.
Q5. Do authors expect an improvement in patients management based on the higher accuracy of their iRCG model?
If our findings are confirmed in larger multi-centre prospective trials, we expect to use the developed metrics to upfront stratify patients by outcome. We have modified the conclusions as:
“Conclusion: CluDiss and the iRCG prognosticated HGSOC outcomes better than conventional and average radiomic measures and could better stratify patient outcomes if validated on larger multi-center trials.”
Introduction:
Q6. 100-102. Authors should clarify if inter-, intra- intra-and-inter-tumor is equal to -site. It might be easier to use site as representative of primary and secondary tumours.
Thank you for the suggestion. We have improved our presentation and consistently used intra- and inter-site radiomic heterogeneity (or IISH) in the paper. This sentence was changed to:
“There is a pressing need for facile and non-invasive measures for intra- and inter-site radiomic heterogeneity that can be integrated into clinical pathways.”,
and
“More recently, we extended these methods to incorporate both intra- and inter-site radiomic heterogeneity (IISH)...”
and
“Furthermore, we developed an integrated marker combining intra- and inter-site radiomics-clinical-genomic (iRCG) variables using machine learning to distinguish patients outcomes.”
Results:
Q7. At the abstract authors stated that there were 75 cases but here it is stated that there were 83 patients and only 44 were used for the radiogenomic analysis and machine learning. This must be clarified if relevant for the iRCG marker. Patients should also be matched with Table 1.
We have now updated Table 1 with the 75 patients used in the study. Only 44 of these had the matched imaging and RNA sequence data and hence the subset analysis of radiogenomic correlation could be performed only that subset. We have now clearly stated this in the manuscript as below:
“After excluding 14 patients who had no platinum resistance data, 61 patients were analyzed for platinum resistance classification. Forty four cases had matched imaging and RNA-sequencing data and used for radiogenomic analysis.”
Q8. The number of images seems not to be relevant. I guess there was 1 CT exam per patient.
Yes. The reviewer is correct. We have removed this sentence.
Q9. The expected improvements in patients’ management by using the iRCG marker should be described.
We have included the following in the results as was previously presented in the discussion:
“Improved prognostication of HGSOC outcomes using the iRCG measures could allow for better upfront and non-invasirve strafication of patients with HGSOC by risk of outcome than with the conventional clinical or genomic measures alone. More accurate risk stratification could faciliate a higher level of intervention for those with the highest risk.”
Discussion:
“Non-invasive stratification of patients with HGSOC by risk of outcome could facilitate a higher level of intervention for those with the highest risk of poor outcome. Possible therapeutic/diagnostic interventions could include enrollment in clinical trials, addition of bevacizumab to first line chemotherapy, and more frequent follow-up imaging to evaluate for progression.”
Q10. How was the robustness to scanner differences calculated if test-retest exams were not performed? This must be clarified (also, 448).
It is indeed true that test-retest analysis is necessary to evaluate repeatability of the radiomics measures to small imaging variations for the same patient undergoing imaging test in the same scanner. However, all the analysis was done retrospectively and we would need to perform a separate study just to assess this, which was not feasible for this study. In addition to repeatability of the measures, the radiomics values also change due to scanner and acquisition variations. Hence, we analyzed whether the radiomic features were robust under different scanners. This is also important as shown in our prior study Um et.al “Impact of image preprocessing no scanner dependence of multi-parametric MRI radiomic features and covariate shift in multi-institutional glioblastoma datasets”, Phys Medicine Biology 2019, because large variabilities between the features can obscure the signal to differentiate between outcomes. We have clarified this in the methods, and included a sentence indicating the need to do further analysis of repeatability in the discussion.
Methods:
“The goal of this experiment was to evaluate whether the variability in the features was due to the different CT scanner manufacturers. This is because large feature variabilities between scanners can obscure the signal to differentiate between the outcomes, thereby reducing the performance of radiomics measures[50].”
Discussion:
“Also, study of repeatability of the radiomic measures needs to be assessed with test-retest studies and under different CT acquisition protocols.”
Discussion:
Q11. Regarding bias, as some radiomic features might vary with different CT protocols, protocols have to be commented (mean values and ranges, contrast doses, acquisition delays).
We have added the following in the discussion:
“The cluDiss measure did not show significant differences to CT scanners, as did 24 of the 75 average heterogeneity measures. However, both first-order histogram and all GLCM measures showed significant difference between scanners.”
and as a limitation the following:
“Also, study of repeatability of the radiomic measures needs to be assessed with test-retest studies and under different CT acquisition protocols.”
We have also added more details of the differences in the CT parameters in the methods:
“Sixty seven patient scans (89%) were acquired with voltage 120kVp (median 120, IQR 110 to 120), tube current (median 300mA, IQR of 89mA to 393mA), and reconstructed with standard convolutional kernel with the most common slice thickness of 5mm for 54 (72%) patients (median 5mm, IQR 2mm to 5mm).”
Materials and Methods:
Q12. 75 patients here. Please, clarify.
We have now clarified this in the text.